# Response Type and Host Species may be Sufficient to Predict Dose-Response Curve Shape for Adenoviral Vector Vaccines

**DOI:** 10.3390/vaccines8020155

**Published:** 2020-03-30

**Authors:** John Benest, Sophie Rhodes, Sara Afrough, Thomas Evans, Richard White

**Affiliations:** 1Department of Infectious Disease Epidemiology, London School of Hygiene and Tropical Medicine, Keppel Street, London WC1E 7HT, UK; sophie.rhodes@lshtm.ac.uk (S.R.); Richard.White@lshtm.ac.uk (R.W.); 2Vaccitech Ltd., The Schrodinger Building, Heatley Road, The Oxford Science Park, Oxford OX4 4GE, UKtom.evans@vaccitech.co.uk (T.E.)

**Keywords:** dosing, dose-response, adenovirus-vectored vaccines, dose dynamics

## Abstract

Vaccine dose-response curves can follow both saturating and peaking shapes. Dose-response curves for adenoviral vector vaccines have not been systematically described. In this paper, we explore the dose-response shape of published adenoviral animal and human studies. Where data were informative, dose-response was approximately five times more likely to be peaking than saturating. There was evidence that host species and response type may be sufficient for prediction of dose-response curve shape. Dose-response curve shape prediction could decrease clinical trial costs, accelerating the development of life-saving vaccines.

## 1. Introduction

Vaccination is effective globally at preventing disease and reducing morbidity and disability [1]. Over recent decades, adenovirus, a vaccine vector used for prophylactic and therapeutic vaccination, has been widely applied, due to both the safety and induction of specific antibodies and T-cells by adenoviral-vectored vaccines [2]. However, adenoviral vaccine developers must still take care to avoid potentially severe adverse effects whilst ensuring that the developed vaccines are efficacious and affordable [3].

A key step in implementing a new vaccine is optimisation of the dosing quantity (hereafter ‘dose’) [4]. As the dose per individual is increased, the cost per individual vaccinated and vaccine toxicity may also increase [5]. We might also assume that the protective efficacy of a vaccine may vary with dose. Optimising a vaccine dose requires establishing a dose that is protective, or induces the highest desired immunological response, whilst avoiding dose-dependent toxicity and minimising cost. Optimising adenoviral vaccines should therefore be approached using multi-objective optimisation methods.

To effectively optimise dose, the relationship between magnitude of dose and immunological response must first be understood. Qualitatively, one might assume that as a dose increases, two types of dose-response relationships may occur, saturating or peaking. A saturating relationship, usually referred to as a sigmoidal response, implies the response is strictly increasing as dose increases, but plateauing so that an increase in dose gives a negligible increase in response beyond a certain threshold. A peaking relationship implies that there exists some dose for which the response is maximised, and that an increase in dose would lead to a decreased response. Historically, pre-clinical trials typically have made the assumption of a saturating curve shape [6]; however, research has shown that for both tuberculosis and influenza vaccines a peaking shape may better describe the dose-response curve [7,8]. Adenoviral vaccine dose-response curve shape has not yet been established.

We aim to provide insight into adenoviral dose-response such that vaccine developers may better optimise adenoviral vector vaccines dosing. Our objectives were:1)assessing the prevalence of peaking/saturating dose-response curve shapes in published adenoviral vector vaccine studies.2)assessing whether dose-response curve shape may be predicted by response type, host species, adenoviral species and route of administration (RoA).3)assessing which of host species, adenoviral species and RoA are the most likely predictors of dose-response curve shape.

Understanding these objectives may be key in predicting likely dose-response curve shape, reducing the number or trial subjects required to determine curve shape and therefore reducing cost for adenoviral dose-response trials.

## 2. Materials and Methods

### 2.1. Data Collation and Preparation

The data were identified in a systematic review of adenoviral dose-response [9], summarised here. In summary, PubMed was searched systematically using terms related to the concepts of adenovirus vector vaccines, immunogenicity, and dose-response through 23 November 2018. Inclusion required prime response data for replication-defective adenovirus vector vaccines with intramuscular (IM) or subcutaneous (SQ) RoA. We excluded cancer models and data where the vaccine was adjuvant coadministered or was recorded post challenge or boost. Non-primate studies were excluded if there were less than 5 animals per dose, and non-human primate data excluded if there were less than 3 individuals per dose. As both representative curves had three unknown parameters, only studies with at least three non-placebo dose-response datapoints were included.

Data collation included vaccine name, vaccine backbone, host species, RoA, response type, response units, time point, whether the response was a summary statistic or individual and the endnote reference for the paper. Studies that provided dose-response data from multiple experiments, and/or multiple time points within the same experiment, were included as separate datasets. In almost all cases the titres were reported as viral particles (VPs), which are less informative in cross study comparisons than infectious units (IUs) [10]. The ratio of VP:IU ranges from 20:1 to 150:1 but is usually found in the 40–80 range.

### 2.2. Objective 1: Assessing the Prevalence of Peaking/Saturating Dose-Response Curve Shapes in Published Adenoviral Vector Vaccine Studies

In this objective we chose representative peaking and saturating curves and calibrated both curves to the dose-response data. Goodness-of-fit tests were used to determine which curve shape best described each trial dataset, and the prevalence of “peaking” or “saturating” curves across the whole dataset was calculated.

#### 2.2.1. Representative Curves

A sigmoidal and gamma probability density function (PDF) were chosen as the representative dose-response curves for saturating and peaking behaviour respectively (Figure 1). See the Appendix B for the curve equations (Equations (A1) and (A2)).

#### 2.2.2. Calibrating Curves to Data

Calibration of the curves to the data was done by finding the parameter estimates that minimise squared error iteratively. Specifically, the nls and nls2 functions in R were used [11,12], which used a brute force algorithm to find reasonable starting parameters and the nl2sol algorithm to further minimise the squared error [13].

The Akaike Information Criterion (AIC) was calculated for both the calibrated peaking and saturating curve for each dataset. The curve shape with the lower AIC was defined as best describing the dataset [14]. The absolute difference between the AIC between peaking and saturating curves for a given dataset was defined as ΔAIC. ΔAIC was used to calculate the support a dataset had for one either curve shape, as follows [14]:

Provides no evidence, ΔAIC <2Positive evidence, 2 ≤ ΔAIC < 6Strong evidence, 6 ≤ ΔAIC < 10Very strong evidence 10 ≤ ΔAIC

A dataset where ΔAIC was greater than or equal to 2 was defined as a dataset “providing evidence” towards either peaking/saturating curve shape and not providing evidence otherwise. Figure 2 shows an example of calibrated peaking and saturating curves and their respective AICs for a three datasets, one for which no curve is superior to the other (ΔAIC <2), and one each for which the peaking/saturating curve was superior (ΔAIC >2)

#### 2.2.3. Calculating Dose-Response Curve Shape Prevalence

The prevalence of datasets providing evidence for either of the curve shapes were calculated. One-sampled, two-tailed t-tests were used to estimate the 95% confidence interval for the probability of a dataset providing evidence for peaking or saturating shape. We also calculated the prevalence of curve shapes by response type, and in only human studies.

#### 2.2.4. Exploring Potential Bias of Independence Assumption

To explore if our conclusions on prevalence of peaking vs saturation curve share were robust to removing datasets from multiple time points with the same study, we repeated the analysis with each study contributing only one dataset.

### 2.3. Objective 2: Assessing whether Dose-Response Curve Shape may be Predicted by Response Type, Host Species, Adenoviral Species, and Route of Administration (RoA)

In this objective we grouped each dataset by response type, host species, adenoviral species and RoA. Within groups, consistency of curve shape was evaluated. This allowed us to assess whether these attributes could predict dose-response curve shape.

#### 2.3.1. Grouping

We grouped each dataset by response type, host species, adenoviral species and RoA.

#### 2.3.2. Evaluating Consistency

This analysis used groups with at least two datasets providing evidence (2 ≤ ΔAIC). A group was defined as consistent if all datasets within that group provided evidence for the same curve shape, and inconsistent if not. A binomial test was used to determine the confidence interval for the probability of within group consistency.

### 2.4. Objective 3: Assessing which of Host Species, Adenoviral Species and RoA are the most likely Predictors of Dose-Response Curve Shape

In this objective we paired datasets that differed in one of the attributes host species, adenoviral species and RoA. Within pairs consistency of curve shape was evaluated. Pairwise consistency was evaluated stratified by the attribute that differed in the pair. This allowed us to assess whether a change in host species, adenoviral species or RoA would lead to a change in dose-response curve shape.

#### 2.4.1. Pairing

Only datasets that provided evidence (2 ≤ ΔAIC) were considered in the analysis. We defined a pair as two datasets with the same response type and two of host species, adenoviral species and RoA the same (Figure 3).

#### 2.4.2. Evaluating Consistency

A pair was defined as consistent if both datasets within that pair provided evidence for the same curve shape, and inconsistent if not. For each of the attributes of host species, adenoviral species and RoA an exact one-sided binomial test was conducted, with trials being the number of pairs for that attribute and successes being the number of consistent pairs for that attribute. This was used to determine the confidence interval for the probability that altering that attribute would not alter the dose-response curve shape. For example, an Antibody/Mouse/Species C/IM dataset could be paired with an Antibody/Mouse/Species C/SQ dataset to examine consistency when varying on RoA.

## 3. Results

### 3.1. Data

The systematic review identified 2787 references, reduced to 581 by title screening, then 300 by screening of the abstract. Screening by full text reduced the number of papers available for analysis to 35 [15,16,17,18,19,20,21,22,23,24,25,26,27,28,29,30,31,32,33,34,35,36,37,38,39,40,41,42,43,44,45,46,47,48,49]. These ranged across five Adenoviral species (B,C,D,E and G), six Host species (Cattle, Human, Monkey, Mouse, Rabbit and Rat), and two routes of administration (IM and SQ). The data ranged across 12 different responses including antibody, T-cell and neutralization titre. The full list of response types considered are in Table 1, and the matrix of papers and their response type/hosts/adenoviral species/RoA can be found in Table A1. Details of the vectors in these papers, including species and origin, can be found in Appendix A.

### 3.2. Objective 1: Assessing the Prevalence of Peaking/Saturating Dose-Response Curve Shapes in Published Adenoviral Vector Vaccine Studies

#### 3.2.1. Overall Prevalence

In total, 191 datasets were extracted from the 35 papers (Appendix A, Appendix A). Datasets on Antibody, T-cell, and CD8+ response data were the most common, each with 30+ datasets. 20+ datasets were available on virus neutralising titre.

Of the 191 datasets, 73.3% (140/191) did not provide evidence for either curve shape (Total that provided evidence, Table 1). Also, 22.0% (42/191) of datasets provided evidence for a peaking shape, and 4.7% (8/191) of datasets provided saturating evidence (total provided evidence = 26.7% (50/191)).

Of datasets that provided evidence for peaking or saturating curve shape, datasets were five times more probable to provide peaking evidence than saturating evidence. Using two-tailed binomial tests with 95% confidence intervals, we estimated that the true probability of a dataset providing peaking evidence across all data was 16.3% to 28.5%, versus 1.8% to 8.1% for saturating evidence.

#### 3.2.2. Prevalence by Response Type

Similarly, the true probability for datasets providing very strong peaking evidence was in the interval 6.9% to 16.3%, compared to 0.0% to 2.9% for very strong saturating evidence [Table 1]. Antibody, T-cell, and Virus Neutralisation responses had datasets providing evidence for both peaking and saturating behaviour. All other responses provided evidence for peaking shape curve shapes only.

#### 3.2.3. Prevalence in Human Data

37 datasets provided data on humans (Table 2). Of these, 56.8% were shown to provide no evidence for either curve shape, 43.2% provided evidence for a peaking shape and 0.0% provided evidence for a saturating shape.

#### 3.2.4. Exploring Potential Bias of Independence Assumption

37 datasets were available after excluding multiple datasets from the same study. Our results were robust to this analysis, with the peaking to saturating evidence ratio remaining approximately 5:1.

### 3.3. Objective 2: Assessing whether Dose-Response Curve Shape may be Predicted by Response Type, Host Species, Adenoviral Species, and Route of Administration (RoA).

52/720 (7.2%) groups contained at least one dataset (Appendix A).

#### Evaluating Consistency

11/52 groups contained at least two datasets that provided evidence (ΔAIC>2) (Table 3). Of these groups 100% (11/11, 95%CI = 71.5–100%, *p* <0.001) were consistent, i.e., all datasets provided evidence for the same curve shape. Of the 11 groups, 18.2% (9/11) only had evidence towards a peaking shape and 81.8% (2/11) only had evidence towards a saturating shape.

### 3.4. Objective 3: Assessing which of Host Species, Adenoviral Species, and RoA are the most likely Predictors of Dose-Response Curve Shape

#### 3.4.1. Evaluating Pairwise Consistency for Host

Of the 50 datasets with evidence, we found 14 pairings such that only the host species was different (Figure 4). 5/14 pairs (27.3%) were consistent. An exact one-sided binomial test with 5 successes in 14 trials gave the 95% confidence interval for the probability of a pairing being consistent as 15.3% to 100.0% with *p*-value = 0.91. This was not considered significant evidence to support predicting curve shape across host species.

#### 3.4.2. Evaluating pairwise consistency for adenoviral species

Of the 50 datasets with evidence, we found 64 pairings such that only the adenoviral species was different (Figure 5), and 60/64 pairs (93.8%) were consistent. An exact one-sided binomial test with 60 successes in 64 trials gave the 95% confidence interval for the probability of a pairing being consistent as 86.3% to 100% with *p*-value < 0.0001. This was considered significant evidence to support predicting curve shape across adenoviral species.

#### 3.4.3. Evaluating pairwise consistency for RoA

Of the 50 datasets with evidence, we found 31 pairings such that only the RoA was different (Figure 6). 30/30 of these pairings (100%) were consistent. A one-sided binomial test with 31 successes in 31 trials gave the 95% confidence interval for the probability of a pairing being consistent as 90.8% to 100%, with *p*-value < 0.001. This was considered significant evidence to support predicting curve shape across RoA.

## 4. Discussion

In this work, peaking and saturating curves were fit to adenoviral data to determine dose-response curve shape overall, and potential adenovirus vaccine attributes were assessed for their potential to determine curve shape. The results suggested that evidence towards a peaked adenoviral dose-response curve occurred five times more frequently than for a saturating curve. Curve shape was consistent within groups, suggesting curve shape was determined by the combined attributes of response type, host species, adenoviral species and RoA. There was strong evidence to support curve shape prediction across adenoviral species and RoA, but not host species.

This study is the first of its kind in the field of adenoviral dose optimization, exploring broad scale patterns in adenoviral dose-response across a large number of different candidate vaccines. The broadness of the data allowed for exploration of the effect of a wide range of vaccine attributes on adenoviral vaccine dose-response, which has not been previously attempted. To our knowledge, a systematic review to extract and evaluate vaccine dose-response data has never been done and this is the first adenoviral example of describing dose-response curve shape.

This work represents the use of an approach towards using quantitative methods to better optimise vaccine dosing. We have presented a rigorous statistical investigation into the prevalence of peaking and saturating curve shapes and into the potential of using vaccine study attributes to predict curve shape. Applying mathematical and statistical methods is novel in adenoviral dose optimisation and was shown to be a reasonable approach towards predicting adenoviral dose response curve shape, a potential advance from more empirical approaches to dose optimisation. Additionally, the calibration of curves and shape analysis did not require specialised software or complex mathematical models, meaning that a similar methodology could be implemented easily in future clinical trials, informed by this work.

There were weaknesses and limitations to our work. Firstly, while we used data covering a broad spectrum of published adenoviral studies, data was relatively sparse, and studies predominantly used mouse hosts and species C adenoviral vectors. Secondly, for 73% of the data, no evidence for either curve shape could be determined. A potential cause for this was that, for some of the datasets, the number of doses analysed was too few or doses too similar in magnitude. It was rare for doses to cross more than three log_10_, which is likely needed to see large differences. Even for datasets where evidence supporting one of the curve shapes was found, further empirical data collection may be useful to confirm the model classifications. Thirdly, it may be possible that neither the simple peaking or saturating shape are optimal to describe adenovirus dose-response curves and there may exist more complex curves that better describe dose-response data for specific groups. However, these simple curves were deemed biologically plausible and chosen to analyse the consistency within and between groups in order to make predictions where possible. Further assessment of curve fitting could be considered if there were enough data sets with greater than 4 points over multiple logs of doses. Fourthly, there was a lot of heterogeneity in the data sets. For example, the extracted data structure was heterogeneous with some datasets on the individual level and others summarised across individuals. The heterogeneity limited the modelling approaches that could be applied to the data.

The methods of this work align well with the new field of Immunostimulation/Immunodynamic (IS/ID) modelling [4]. This field suggests modelling as a useful approach towards understanding how a vaccine’s efficacy depends on dose. Similar to our work, other IS/ID modelling studies have fit statistical curves and mathematical models to dose-response data for novel tuberculosis vaccine, H56+IC31 [8] and parainfluenza/influenza vaccines HPIV-3 and IAV [7] to determine curve shape. For H56+IC31 and HPIV-3, peaking curve shape was found to be a better description of the dose-response curve shape compared to a saturating shape [7,8], consistent with our results for adenoviral vaccines. However, the influenza vaccine, IAV, was found to be saturating [7].

We found that for adenoviral vaccines, peaking dose-response curve shapes are more prevalent than saturating shapes. Dose escalation studies may therefore not be an appropriate method of adenoviral dose optimisation, as they typically assume a saturating dose-response curve shape and select the maximal safe dose. As such, we recommend that adenoviral dose-response studies should include analysis of immunogenic outcomes chosen by their likelihood of association with protective responses, and not just toxicology, to optimise dose.

In objectives 2 and 3 we showed it may be possible to predict adenoviral dose-response curve shape using data from similar adenoviral dose-response studies. We suggested the likely attributes that would need to be the same to justify that prediction, i.e., response type, host species, adenoviral species, and RoA. We also showed that host species and response type may be sufficient for prediction of dose-response curve shape. Hence, if the dose-response curve shape is known for one vaccine for a given host species and response type, we may be able to inform and predict dose response curve shape for another vaccine regardless of differences in adenoviral species and RoA.

We could not explore the effects of other potential attributes that could influence dose-response curve shape. Stratifying by pathogen would have reduced group size, preventing analysis of curve shape within group. There exist other attributes that some of the data did not measure, which could similarly not be used to establish impact on dose response curve shape. Whilst this is a limitation of our work, we still found groups to be 100% consistent. This may imply that attributes that were not explored in this work are not required for predicting curve shape. This would support prediction of dose-response curve shape using dose-response data from vaccine targeted against a different pathogen. We did not stratify by whether adenovirus vectors were simian or human derived. This could be future areas of adenoviral dose-response curve research.

Though we have shown that prediction of adenoviral vaccine dose-response may be possible, validating such predictions would be an important area of future research. Which attributes cause the shape to be peaking and which cause the shape to be saturating should also be determined, as we assessed consistency of shape not the causal pathway in which attributes may alter dose response curve shape. Understanding the attribute causation of curve shape could result in the development of a tool that can suggest the probability that a novel vaccine will have a peaking or saturating curve shape. Aggregation of future adenoviral dose-response studies with those analysed here would aid in developing such a tool. Even without such a tool, this work suggests that to predict curve shape all that is required is to have a previously established adenoviral dose response curve shape for that response type and species.

Whilst this work showed evidence that vaccine dose-response curve shapes may be either peaking or saturating, a method of optimally designing vaccine trials to best determine dose-response curve shape is yet unknown. This is another area where future research could lead to better dose optimisation.

Statistical models, like those in this paper, allow shape to be determined. With sufficient data, a mechanistic model, like those used in [7,8], may allow for a more accurate prediction of optimal dose and deeper understanding of longitudinal dose-response, and such approaches merit further investigation in an adenoviral context.

## 5. Conclusions

We found that where data were informative, dose-response was approximately five times more likely to be peaking than saturating. We also found that there was evidence that host species and response type may be sufficient for prediction of dose-response curve shape. We found that where data were informative, dose-response was approximately five times more likely to be peaking than saturating. We also found that there was evidence that host species and response type may be sufficient for prediction of dose-response curve shape.

## Figures and Tables

**Figure 1 vaccines-08-00155-f001:**
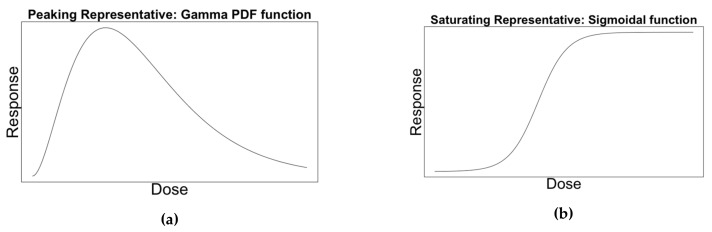
Examples of a representative curve for (**a**) peaking and (**b**) saturating behaviour.

**Figure 2 vaccines-08-00155-f002:**
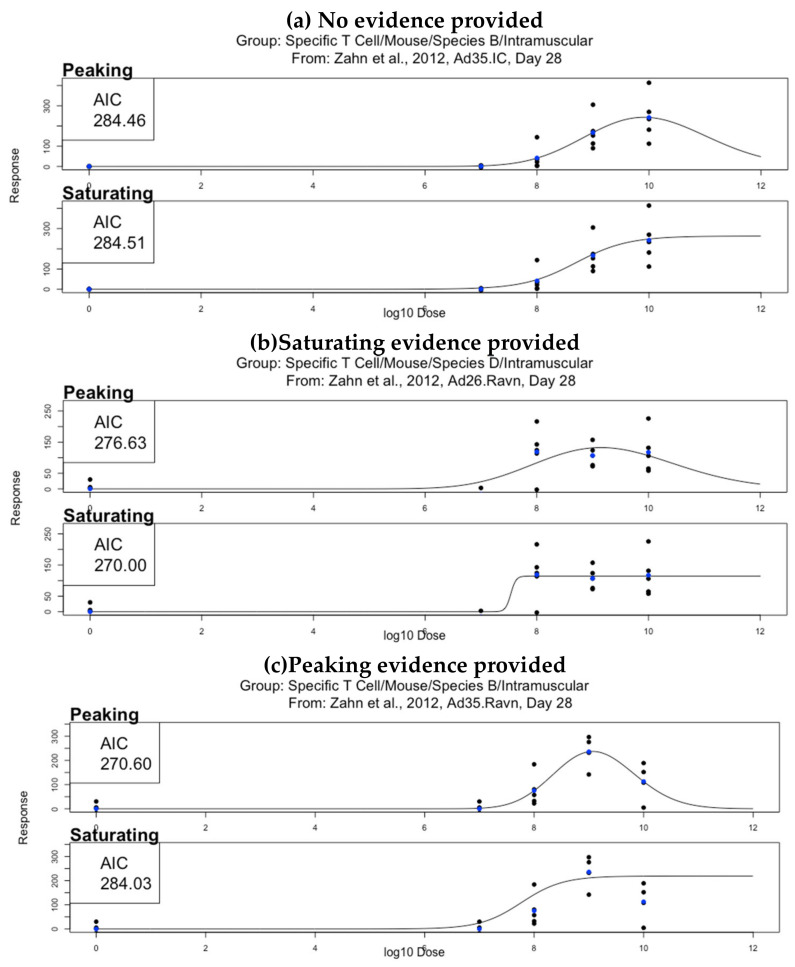
An example of three datasets; (**a**) does not provide evidence to support one curve shape over the other (i.e., ΔAIC = 0.05 < 2), (**b**) provides evidence to support a saturating curve shape over a peaking curve shape (i.e., ΔAIC = 6.63 > 2), and (**c**) provides evidence to support a peaking curve shape over a saturating curve shape (i.e., ΔAIC = 13.43 > 2). For each dataset, the top plot shows the calibrated peaking curve and associated Akaike Information Criterion (AIC), and the bottom plot shows the calibrated saturating curve and associated AIC. Black dots represent individual mice and blue dots represent the mean response for that dose.

**Figure 3 vaccines-08-00155-f003:**
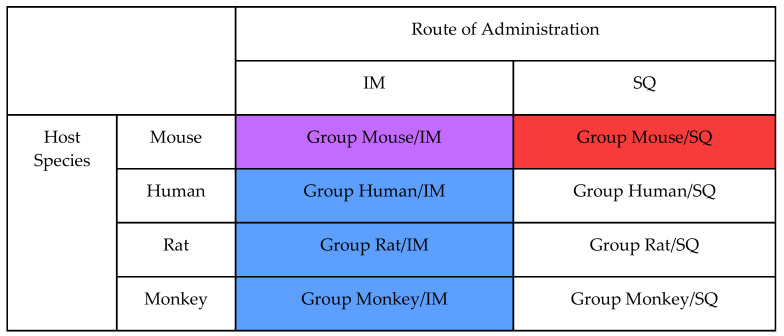
A representation of the definition of a comparing across group, simplified to the 2 attributes of host and route of administration (RoA). For analysis of consistency of shape across groups, a dataset from group Mouse/IM (purple cell) would be paired with only datasets from group Mouse/SQ when predicting across RoA (red cells), and with only datasets from groups Human/IM, Rat/IM and Monkey/IM when predicting across host species (blue cells). This was extended to the full 4 attribute dimensions in the analysis.

**Figure 4 vaccines-08-00155-f004:**
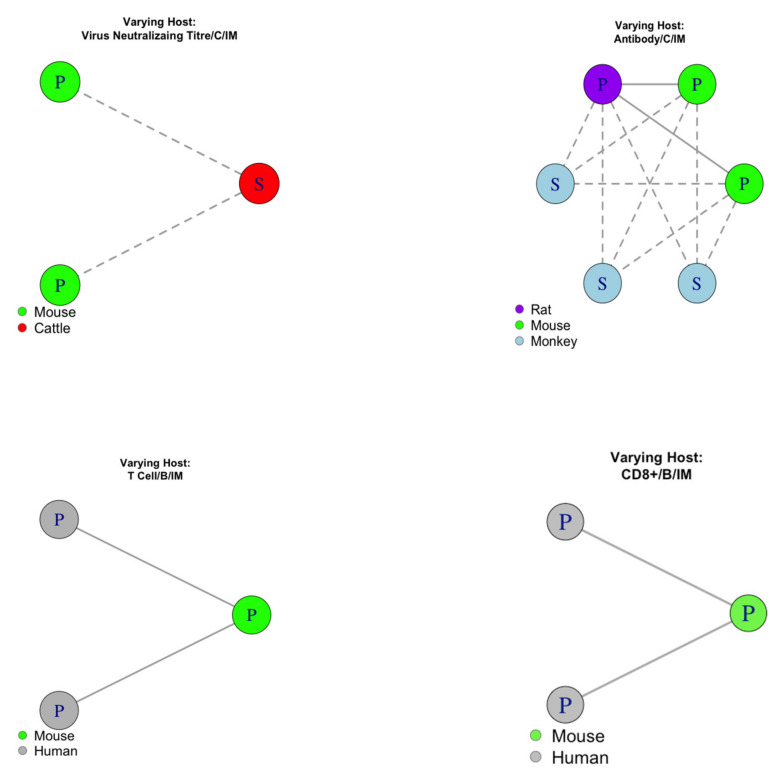
Pairings that differed in host species, examined for analysis of consistency between groups. Each point represents a dataset that provided evidence, where P/S denotes the dataset had evidence towards a peaking/saturating shape, respectively. Datasets were colour coded according to their host. A solid line shows consistent pairings and a dashed line shows inconsistent pairings. B = species B adenoviral vector. C = species C adenoviral vector. IM = Intramuscular route of administration.

**Figure 5 vaccines-08-00155-f005:**
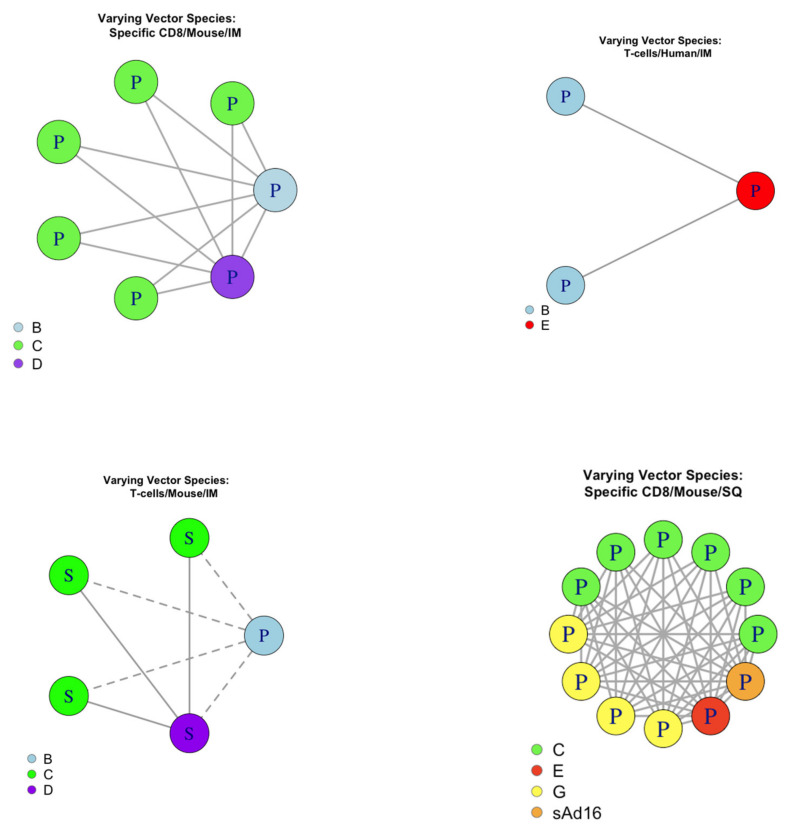
Pairings that differed in host species examined for analysis of consistency between groups. Each point represents a dataset that provided evidence, where P/S denotes the dataset had evidence towards a peaking/saturating shape, respectively. Datasets were colour coded according to their vector species. A solid line shows consistent pairings and a dashed line shows inconsistent pairings. B = species B adenoviral vector. C = species C adenoviral vector. D = species D adenoviral vector. E = species E adenoviral vector. G = species G adenoviral vector. sAd16 = adenoviral vector was sAd16. IM = Intramuscular route of administration. SQ = Subcutaneous route of administration.

**Figure 6 vaccines-08-00155-f006:**
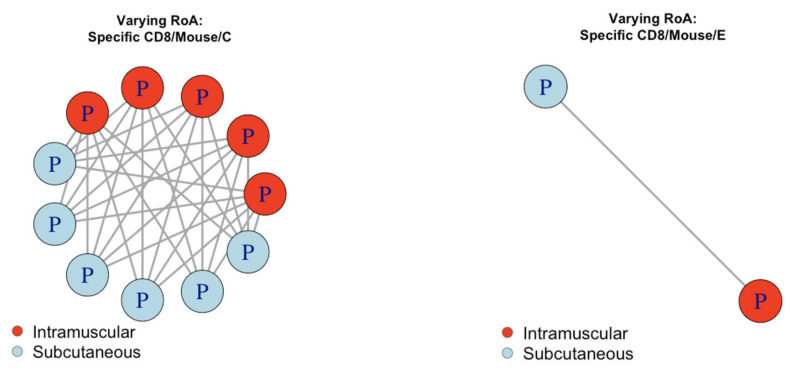
Pairings that differed in RoA species examined for analysis of consistency between groups. Each point represents a dataset that provided evidence, where P/S denotes the dataset had evidence towards a peaking/saturating shape, respectively. Datasets were colour coded according to their Route of Administration. A solid line shows consistent pairings and a dashed line shows inconsistent pairings. C = species C adenoviral vector. E = species E adenoviral vector.

**Table 1 vaccines-08-00155-t001:** Number of datasets for each level of evidence, sorted by response type.

	Total Number of All Datasets by Level of Evidence and Response							
Response Type	Very Strong Peaking 10≤ ΔAIC	Strong Peaking 6≤ ΔAIC <10	Positive Peaking 2≤ ΔAIC <6	No Evidence ΔAIC<2	Positive Saturating 2≤ ΔAIC <6	Strong Saturating 6≤ ΔAIC <10	Very Strong Saturating 10≤ ΔAIC	Total Number of Datasets for Response
Antibody	3	0	2	46	1	1	1	54
T-Cell	1	2	0	27	3	1	0	34
CD4+	1	1	0	4	0	0	0	6
CD8+	11	3	7	42	0	0	0	63
CD4 IFNγ+	1	1	0	0	0	0	0	2
CD8 IFNγ+	1	0	0	2	0	0	0	3
CD4+ TNFα+	0	0	1	1	0	0	0	2
CD8+ TNFα+	0	0	1	0	0	0	0	1
CD4+ IL-2+	1	0	0	1	0	0	0	2
CD8+ IL-2+	0	0	1	0	0	0	0	1
CD4+ IL-17+	0	0	1	0	0	0	0	1
Virus Neutralization Titre	2	0	1	17	1	1	0	22
Total that provided evidence (%)	42 (22.0%)			140 (73.3%)	8 (4.7%)			191

The far left column indicates response for that row type, the far right column gives the response type distribution of the whole dataset. The other columns indicate type of evidence and the number of datasets found for that response type and type of evidence. The opacity of the red colouring was equal to the proportion of datasets for that row’s response type for which that type of evidence was found, so a darker red indicates that for this response type this type of evidence was more prevalent than for a cell in that row with a lighter red. The opacity of the blue colouring for each response type in the far right column was equal to the proportion of total datasets which had that response type, so a darker blue indicates that this response type was more prevalent in the whole dataset than for a cell with a lighter blue. The final row summarises the number of datasets that provided peaking evidence, saturating evidence, or no evidence across all datasets.

**Table 2 vaccines-08-00155-t002:** Number of datasets for each level of evidence, for human only data.

	Total Number of Human Datasets by Associated Level of Evidence						
	Very Strong Peaking 10≤ΔAIC	Strong Peaking 6≤ΔAIC<10	Positive Peaking 2≤ΔAIC<6	No Evidence ΔAIC<2	Positive Saturating 2≤ΔAIC<6	Strong Saturating 6≤ΔAIC<10	Very Strong Saturating 10≤ΔAIC
All responses (%)	6 (16.2%)	4 (10.8%)	6 (16.2%)	21 (56.8%)	0 (0%)	0 (0%)	0 (0%)
Total that provided evidence (%)	16 (43.2%)			21 (56.8%)	0 (0%)		

The opacity of the red colouring was proportional to the percentage of datasets that with that associated level of evidence, so a darker red indicates that this type of evidence was more prevalent than for a cell in that row with a lighter red.

**Table 3 vaccines-08-00155-t003:** Summary of consistency of curve shape within groups.

Group (Response Type /Host/Adenoviral Species/Route of Administration)	Number of Datasets Providing Evidence Towards a Peaking Shape	Number of Datasets Providing Evidence Towards a Saturating Shape	Consistency
Antibodies/Human/B/IM	2	0	Consistent
Antibodies/Mouse/C/IM	2	0	Consistent
Antibodies/Monkey/C/IM	0	3	Consistent
T Cell/Mouse/C/IM	0	3	Consistent
CD4+/Human/B/IM	2	0	Consistent
CD8+/Human/B/IM	2	0	Consistent
CD8+/Mouse/C/IM	5	0	Consistent
CD8+/Mouse/C/SQ	6	0	Consistent
CD8+/Mouse/G/SQ	4	0	Consistent
CD4+ IFNγ+/Human/B/IM	2	0	Consistent
Virus Neutralization Titre/Mouse/C/IM	2	0	Consistent

B = species B adenoviral vector. C = species C adenoviral vector. G = species G adenoviral vector. IM = intramuscular RoA. SQ = subcutaneous RoA.

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
