# Peer review of "Response Type and Host Species may be Sufficient to Predict Dose-Response Curve Shape for Adenoviral Vector Vaccines"

_vaccines, 2020, doi:10.3390/vaccines8020155_

Round 1

Reviewer 1 Report

The premise of this study is that the dose-response curve for an adenovirus-vectored vaccine describes a curve with a peak, which indicates an optimum response at less than maximal dose. This pattern is different from the typical expectation of a saturating response with an S-shaped response over physiological ranges of the vaccine. Presumable, at supra-physiological levels any biological response would be diminished. This work performed a meta-analysis of published studies by using non-linear regression to determine if the data are better described by one curve or another and then attempted to determine if the shape of the curve can be predicted by various test variables. The criteria for discriminating between models was Akaike's "An Information Criterion" that provides a log-likelihood value for the goodness of fit adjusted for the number of parameters in the fitted model.

The study is intriguing and rather ambitious. The ambitious nature of this study is due to the requirement of obtaining sufficient data points spanning a meaningfully wide range of doses in published studies. Nonetheless, of 219 data sets satisfying the screening criteria, 25% provide evidence in favor of either curve shape. Among these, 20% provided evidence for a peaked curve (gamma function) with only 5% favoring a sigmoidal curve. This meta-analysis suggests that the dose-response is more frequently described by a gamma distribution (peaking) than sigmoidal distribution. This study points to the importance of considering this property in the design of a dose escalation studies for adenovirus-vectored vaccines. The results described here provide a novel view of vaccine dose-response studies that should inform future trials and may provoke additional studies to examine the biological basis for potential properties unique to adenovirus vectors.

The results are necessarily limited by the availability of published data and are consequently largely descriptive and limited in scope. Nonetheless, the trends uncovered here appear scientifically sound and justified. A few concerns have been identified in the manuscript which should be considered by the authors.

  1. The methodology under Objective 3, which sought to assess the degree to which of the host species, adenovirus species and route of administration could predict does-response was not well described. The document mentions a pairwise comparisons stratified by the attribute in question but the statistical tests used to discern significant outcomes was not clearly described.
  2. The text in the legend to Fig. 6 seems a bit fragmented and could be revised for clarity.
  3. As noted by the authors, the curve-fitting algorithm was limited to one of two analytic models. For this reason, it was not surprising that the majority of studies failed to yield a favored form. I wonder if a less parameterized, non-closed equation could have yielded more information with a Bayesian analysis? This may be a viable alternative to using a more complex curve that could better describe the dose-response at the cost of additional parameters as suggested in the Discussion. This may be beyond the scope of this exploratory study but is offered for consideration by the authors.
  4. Although the entire data set with fitted curves is to be available as supplemental information, it would have been helpful to include representative data and curves showing a "peaking" distribution as well as a "sigmoidal" distribution in addition to the indeterminate data shown in Fig. 2.

Author Response

The premise of this study is that the dose-response curve for an adenovirus-vectored vaccine describes a curve with a peak, which indicates an optimum response at less than maximal dose. This pattern is different from the typical expectation of a saturating response with an S-shaped response over physiological ranges of the vaccine. Presumable, at supra-physiological levels any biological response would be diminished. This work performed a meta-analysis of published studies by using non-linear regression to determine if the data are better described by one curve or another and then attempted to determine if the shape of the curve can be predicted by various test variables. The criteria for discriminating between models was Akaike's "An Information Criterion" that provides a log-likelihood value for the goodness of fit adjusted for the number of parameters in the fitted model.

The study is intriguing and rather ambitious. The ambitious nature of this study is due to the requirement of obtaining sufficient data points spanning a meaningfully wide range of doses in published studies. Nonetheless, of 219 data sets satisfying the screening criteria, 25% provide evidence in favor of either curve shape. Among these, 20% provided evidence for a peaked curve (gamma function) with only 5% favoring a sigmoidal curve. This meta-analysis suggests that the dose-response is more frequently described by a gamma distribution (peaking) than sigmoidal distribution. This study points to the importance of considering this property in the design of a dose escalation studies for adenovirus-vectored vaccines. The results described here provide a novel view of vaccine dose-response studies that should inform future trials and may provoke additional studies to examine the biological basis for potential properties unique to adenovirus vectors.

The results are necessarily limited by the availability of published data and are consequently largely descriptive and limited in scope. Nonetheless, the trends uncovered here appear scientifically sound and justified. A few concerns have been identified in the manuscript which should be considered by the authors.

 We thank the reviewer for their time in reviewing this manuscript.  We have updated the manuscript and supplementary in response to comments from all reviewers, and will upload these if invited to resubmit. 

The methodology under Objective 3, which sought to assess the degree to which of the host species, adenovirus species and route of administration could predict does-response was not well described. The document mentions a pairwise comparisons stratified by the attribute in question but the statistical tests used to discern significant outcomes was not clearly described.

Thank you for this comment.  We have added to the description of that method to clarify the statistical approach: “an exact 1 sided binomial test was conducted, with trials being the number of pairs for that attribute and successes being the number of consistent pairs for that attribute.”

The text in the legend to Fig. 6 seems a bit fragmented and could be revised for clarity.

We thank you for this comment. We have updated the legend of figure 6 to include unabbreviated routes of administration. We have also rewritten the captions for figures 4-6 to clarify further. 

As noted by the authors, the curve-fitting algorithm was limited to one of two analytic models. For this reason, it was not surprising that the majority of studies failed to yield a favored form. I wonder if a less parameterized, non-closed equation could have yielded more information with a Bayesian analysis? This may be a viable alternative to using a more complex curve that could better describe the dose-response at the cost of additional parameters as suggested in the Discussion. This may be beyond the scope of this exploratory study but is offered for consideration by the authors.

Thank you for this interesting suggestion. Our primary question was whether the data better support a peaking or saturating curve, and, as such, this required us to use peaking or saturating curve shapes. However, as suggested, in future, if we were interested in identifying any curve shape we would use alternative methods.

Although the entire data set with fitted curves is to be available as supplemental information, it would have been helpful to include representative data and curves showing a "peaking" distribution as well as a "sigmoidal" distribution in addition to the indeterminate data shown in Fig. 2.

Thank you for this excellent suggestion. It has been be added to figure 2, and figures from the supplementary reformatted to be appropriate to the main text. Please see the attachment.

Reviewer 2 Report

In this manuscript, Benest et al. perform a literature review of studies using adenovirus vaccine vectors. Using that data, they attempt to determine if adenovirus vectors generate saturating curves or peaked curves. The authors calculate that most studies have peaking curves (Table 1). However, many data in the supplemental material have so few points that is it difficult to determine if that is a conclusion. Overall the manuscript is poorly organized and difficult to follow. The line numbers have been removed from the template. This makes it extremely difficult for review. The authors need to reorganize the manuscript and present the data in a clear way (see specifics below) so that this reviewer and readers can follow the data.

The abstract should state the results of the study (peak vs. saturation vs. it could not be determined)

The supplemental material is difficult to follow. There are no Figure or Table legends. There are also multiple pages with numbers but no figures. Many figures have “Response” as the y-axis title with no units. Therefore, this reviewer cannot follow which data are used. It is noted that most figures are generated off of 3-4 data points, which are not enough to determine saturation vs. peak and this reviewer cannot determine which curves are used for which data (the authors point out that 75% of the data could not determine curve shape).

Section 2.1: “submitted” and then referencing a paper is not appropriate citations. This is unpublished data.

Objectives should not be listed as Materials and Methods. The titles of the methods should only list the methods used. Same with 3.1 (which is currently labeled “Data”). Many methods and results can be combined instead of separate sections (Bias was calculated…).

The authors reference the supplemental instructions for equations [Equations A1]. There are no equations in the supplemental instructions and they are found in Appendix A.

Table 2: Red coloring is in areas that % is not calculated.

Author Contributions: Please remove the instructions.

Table 1 in Appendix A: citations do not need a separate table.

Author Response

In this manuscript, Benest et al. perform a literature review of studies using adenovirus vaccine vectors. Using that data, they attempt to determine if adenovirus vectors generate saturating curves or peaked curves. The authors calculate that most studies have peaking curves (Table 1). However, many data in the supplemental material have so few points that is it difficult to determine if that is a conclusion. Overall the manuscript is poorly organized and difficult to follow. The line numbers have been removed from the template. This makes it extremely difficult for review. The authors need to reorganize the manuscript and present the data in a clear way (see specifics below) so that this reviewer and readers can follow the data.

We thank the reviewer for their time in reviewing this manuscript.  We have already updated the manuscript and supplementary in response to comments from all reviewers, and will upload these if invited to resubmit. 

The abstract should state the results of the study (peak vs. saturation vs. it could not be determined)

We thank the reviewer for suggesting this important clarification. We have replaced the following text in the abstract "Dose-response was five times more likely to be peaking than saturating." with “Where data were informative, dose-response was five times more likely to be peaking than saturating.

The supplemental material is difficult to follow. There are no Figure or Table legends. There are also multiple pages with numbers but no figures. Many figures have “Response” as the y-axis title with no units. Therefore, this reviewer cannot follow which data are used. It is noted that most figures are generated off of 3-4 data points, which are not enough to determine saturation vs. peak and this reviewer cannot determine which curves are used for which data (the authors point out that 75% of the data could not determine curve shape).

Thank you for this useful comment. We have restructured the supplementary material to incorporate these comments as follows:

  1. We have added preamble to the supplementary to better explain the structure of this document. 
  2. The reviewer is correct that the lower half of the supplementary appears to have been lost. We apologise. This has been fixed in the present supplementary PDF.
  3. We have updated the supplementary to have the name of the paper included above each dataset. This is to make it easier to locate where the data came from so the reader is able to determine which units were used. Additionally, the supplementary preamble now explains that datasets are listed under the response type.
  4. In the supplementary preamble, we have included text to specify that, for each dataset, plots on the left of the page show the data overlaid with the calibrated saturating curve, and those on the right side show the same data overlaid with the calibrated peaking curve. All datasets include a plot for both curve shapes. 

Section 2.1: “submitted” and then referencing a paper is not appropriate citations. This is unpublished data.

Thank you for this comment. This has been updated to “unpublished data, personal communication, Afrough”. If this paper is accepted before this manuscript, this statement will be updated to include a full reference. We also have uploaded this in-review manuscript in case the reviewers would like to view it. Please see the attachment.

Objectives should not be listed as Materials and Methods. The titles of the methods should only list the methods used. Same with 3.1 (which is currently labeled “Data”). Many methods and results can be combined instead of separate sections (Bias was calculated…).

Thank you for this comment. This method structure was deliberate and allows the specific methods used for each objective to be crystal clear and reproducible.  We believe the paper is much clearer if separate methods sections are reported for each objective, and therefore have not made any changes in response to this comment. However if the editor prefers the reviewer’s suggested style we would be happy to rewrite this section.

The authors reference the supplemental instructions for equations [Equations A1]. There are no equations in the supplemental instructions and they are found in Appendix A.

Thank you. The text has been changed to reflect the equations in the appendix, not the supplementary. 

Table 2: Red coloring is in areas that % is not calculated.

Thank you. Percentages have been added to these cells.

Author Contributions: Please remove the instructions.

Thank you. This has been removed in the resubmitted version.

Table 1 in Appendix A: citations do not need a separate table.

Thank you. The tables have been combined to include the reference alongside the paper number.

Reviewer 3 Report

In this manuscript, Benest et al compare two models of immune dynamics, saturating and peaking. The authors model the immune response from 35 different studies in animal models and humans and graph antibody responses and T cell responses.  The authors then fit the data to a peaking curve or a saturating curve and identify which model fits the data better. The authors find that most data show no evidence of a peaking or saturating immune dynamics. But the immune response is more likely represents peaking dynamics compared to a saturating dynamics.

However the central premise of this manuscript is flawed as the modeling techniques cannot discriminate between the two types of immune response dynamics. In many examples the immune response peaks in a dose dependent manner, and stays at the peak response with higher doses, representative of a classical saturating immune response. Examples include samples 936-ADHA wt Ab Week 5, 2531-gD Ab D10, 1539 AdC7PA-14dpi, 576 Ad35-PBMC, 578 Ad26-Angola, and 578 Ad26-S/G. However, the data can also fit onto a peaking curve, with a similar AIC. Thus the modeling is biased to fit the data to a peaking curve, even when the data represents saturating dynamics.

Also pages 37-81 of the supplementary data were blank thus a large portion of the data is missing.

Author Response

In this manuscript, Benest et al compare two models of immune dynamics, saturating and peaking. The authors model the immune response from 35 different studies in animal models and humans and graph antibody responses and T cell responses.  The authors then fit the data to a peaking curve or a saturating curve and identify which model fits the data better. The authors find that most data show no evidence of a peaking or saturating immune dynamics. But the immune response is more likely represents peaking dynamics compared to a saturating dynamics.

However the central premise of this manuscript is flawed as the modeling techniques cannot discriminate between the two types of immune response dynamics. In many examples the immune response peaks in a dose dependent manner, and stays at the peak response with higher doses, representative of a classical saturating immune response. Examples include samples 936-ADHA wt Ab Week 5, 2531-gD Ab D10, 1539 AdC7PA-14dpi, 576 Ad35-PBMC, 578 Ad26-Angola, and 578 Ad26-S/G. However, the data can also fit onto a peaking curve, with a similar AIC. Thus the modeling is biased to fit the data to a peaking curve, even when the data represents saturating dynamics.

We thank the reviewer for their time in reviewing this manuscript. We have already updated the manuscript and supplementary in response to comments from all reviewers, and will upload these if invited to resubmit. 

We believe we may not have been clear enough and therefore the reviewer may have misunderstood the methods and conclusions of the analysis. In the majority of cases highlighted by the reviewer, our methods correctly identified that it could not be determined whether the dose-response curve shape was peaking or saturating. 

As an example, for 936-ADHA wt Ab Week 5, the data does seem to show response increasing with dose. However, our methods were unable to distinguish whether a further increased dose would lead to an increased, equal, or decreased response. 

This is a direct consequence of the lack of dosing data, and, as we report, suggests that without testing higher doses, a peaking or saturating dose response curve shape cannot be determined for many datasets.

We appreciate this was not as clear as it could be. Therefore we have added illustrative plots to figure 2. Please see the attachment.

Also pages 37-81 of the supplementary data were blank thus a large portion of the data is missing.

We apologise for the oversight. This has been fixed in the present supplementary PDF.

Reviewer 4 Report

Dose-response studies are an important aspect during the development of new vaccines. They are conducted to evaluate the safety and to determine the optimal efficacy. In general, vaccine dose-response curves can follow both saturating and peaking shapes. In this study the authors concentrated on adenoviral vector vaccines and aimed to provide insight into adenoviral dose-response shapes. Specifically, the authors wanted to know whether it is possible to predict likely dose-response curve shapes based on different parameters such as response type, host species, adenoviral species and route of administration (RoA). Predictability of dose-response shapes based on these parameters could lead to a reduction in the number of trial subjects required to determine the curve shape, and therefore could reduce cost for adenoviral dose-response trials. The authors conducted a systematic review of existing literature for adenoviral vaccine dose -response studies. Only 35 studies were identified that were available for statistical analysis. They ranged across four Adenoviral species, six Host species, two routes of administration and there were 15 different responses measured. The authors extracted 219 data sets from these 35 papers. Notably, 75.3% of these did not provide evidence for either curve shape. 20.5% of datasets provided evidence for a peaking type and only 4.1% of datasets provided saturating evidence. Thus, one of the major findings of this analysis is that datasets were five times more probable to provide peaking evidence than saturating evidence. The data provide some evidence that curve shape is determined by the combined attributes of response type, host species, adenoviral species, and RoA. The authors found strong evidence to support curve shape prediction across adenoviral species and RoA, but not host species.

The study is very well executed and the manuscript is well written. The analysis is the first of its kind and might potentially be very helpful for vaccine developers. However, as the authors are very well aware of and as pointed out in the discussion, there are many limitations to their study and therefore, at it stands right now, I cannot see that there is an immediate large benefit for new vaccine study designs employing adenovirus vectored vaccines. Obviously, due to the relatively small number of studies amenable to statistical analysis, the conclusions need to be validated based on a larger database. This is complicated by the fact that the vast majority of the studies actually did not provide evidence for either curve shape. As the authors discuss, due to the limited number of datasets, it was not possible to stratify the data based on pathogen. This in my opinion is a major caveat since it seems quite conceivable that even with all other parameters being identical, the nature of the pathogen might dictate the shape of the dose-response curve. Therefore, it remains to be seen whether predictions can be reliably made with respect to dose-response curves. Furthermore, while the authors stratified the data for adenovirus species based on four subgroups, another important characteristic to look at is whether the adenovirus vectors were all replication incompetent or replication competent. Would this change the outcome of the statistical analysis? Furthermore, some of the adenoviral vectors are of simian / chimpanzee origin in order to mitigate the effect of existing immunity to human adenovirus derived vectors. This also should be taken into consideration. Information on the exact subtype of Adenoviral vector and whether it is replication competent or not should be provided as supplemental information.

Minor comments:

Page 6: “ , and 4.1% (9/219) datasets provided …” Please change to “, and 4.1% (9/219) of datasets provided….”

Page 11: “… and influenza vaccines HPIV-3 and IAV ….”: Please note that HPIV-3 is not an influenza virus- it is human parainfluenza virus and the disease it causes is not influenza.

Page 12: “We also showed that that host species and response type …..”. Typo: Please delete the 2nd “that”.

Author Response

Dose-response studies are an important aspect during the development of new vaccines. They are conducted to evaluate the safety and to determine the optimal efficacy. In general, vaccine dose-response curves can follow both saturating and peaking shapes. In this study the authors concentrated on adenoviral vector vaccines and aimed to provide insight into adenoviral dose-response shapes. Specifically, the authors wanted to know whether it is possible to predict likely dose-response curve shapes based on different parameters such as response type, host species, adenoviral species and route of administration (RoA). Predictability of dose-response shapes based on these parameters could lead to a reduction in the number of trial subjects required to determine the curve shape, and therefore could reduce cost for adenoviral dose-response trials. The authors conducted a systematic review of existing literature for adenoviral vaccine dose -response studies. Only 35 studies were identified that were available for statistical analysis. They ranged across four Adenoviral species, six Host species, two routes of administration and there were 15 different responses measured. The authors extracted 219 data sets from these 35 papers. Notably, 75.3% of these did not provide evidence for either curve shape. 20.5% of datasets provided evidence for a peaking type and only 4.1% of datasets provided saturating evidence. Thus, one of the major findings of this analysis is that datasets were five times more probable to provide peaking evidence than saturating evidence. The data provide some evidence that curve shape is determined by the combined attributes of response type, host species, adenoviral species, and RoA. The authors found strong evidence to support curve shape prediction across adenoviral species and RoA, but not host species.

The study is very well executed and the manuscript is well written. The analysis is the first of its kind and might potentially be very helpful for vaccine developers. However, as the authors are very well aware of and as pointed out in the discussion, there are many limitations to their study and therefore, at it stands right now, I cannot see that there is an immediate large benefit for new vaccine study designs employing adenovirus vectored vaccines. Obviously, due to the relatively small number of studies amenable to statistical analysis, the conclusions need to be validated based on a larger database. This is complicated by the fact that the vast majority of the studies actually did not provide evidence for either curve shape. As the authors discuss, due to the limited number of datasets, it was not possible to stratify the data based on pathogen. This in my opinion is a major caveat since it seems quite conceivable that even with all other parameters being identical, the nature of the pathogen might dictate the shape of the dose-response curve. Therefore, it remains to be seen whether predictions can be reliably made with respect to dose-response curves. Furthermore, while the authors stratified the data for adenovirus species based on four subgroups, another important characteristic to look at is whether the adenovirus vectors were all replication incompetent or replication competent. Would this change the outcome of the statistical analysis? 

We thank the reviewer for highlighting that this paper is ‘the first of its kind’.  We have already updated the manuscript and supplementary in response to comments from all reviewers, and will upload these if invited to resubmit. 

In response to the question of replication. We agree that the dose-response dynamics of replicating compared to replication incompetent may differ. Therefore one of the inclusion criteria was that the adenovirus vectors must be not replicating. See line 65 of the revised manuscript. 

Furthermore, some of the adenoviral vectors are of simian / chimpanzee origin in order to mitigate the effect of existing immunity to human adenovirus derived vectors. This also should be taken into consideration. 

The reviewer's suggestion of including whether the vectors were derived from human or simian adenovirus was excellent. We have added a new section in the supplementary that describes the vectors, their species, and whether they are human derived. These additions have been included as a pdf alongside this response, and are labelled S1a and S1b in the updated supplementary. Please see the attachment.

For objective 2, stratifying on this adenovirus origin does not change the outcome, as groups were fully consistent already. If we had observed inconsistency in group dose-response curve shape in objective 2, the possibility that curve shape may be dependent on adenovirus origin would have been stated as a potential cause. Stratifying by origin would have also reduced group size, which causes the same issues with stratifying by pathogen, noted on lines 363-364.

For objective 3, we considered the pairings that were found to be inconsistent and for which the vector used in one of the datasets in the pair was human-derived and the vector used in the other dataset was simian-derived. For the analysis of all three attributes of RoA, Host, and Adenoviral species, there were not enough of such pairings that any statistical analysis could be carried out to examine the possibility that the inconsistencies were due to differences in the origin of the vector. Therefore, this analysis has not been added to the paper.

Overall, this is an excellent question, but we need more empirical data to be able to answer it. We have added the following text to the conclusion to highlight the potential importance of this question; “We did not stratify by whether adenovirus vectors were simian or human derived. This could be future areas of adenoviral dose-response curve research.”

Information on the exact subtype of Adenoviral vector and whether it is replication competent or not should be provided as supplemental information.

We again thank the reviewer, and include a pdf of the additions made to the supplementary. Please see the attachment.

Minor comments:

Page 6: “ , and 4.1% (9/219) datasets provided …” Please change to “, and 4.1% (9/219) of datasets provided….”

Page 11: “… and influenza vaccines HPIV-3 and IAV ….”: Please note that HPIV-3 is not an influenza virus- it is human parainfluenza virus and the disease it causes is not influenza.

Page 12: “We also showed that that host species and response type …..”. Typo: Please delete the 2nd “that”.

We thank reviewer 4 for their detailed and positive feedback on this work. We have included all revisions in the updated manuscript.

Round 2

Reviewer 2 Report

Vaccine-726429-v2

Thank you for including the author’s response to reviewers to aid in review as well as track changes and highlights. The authors have made extensive changes to the supplemental material to make it much easier to read and follow (as much as can be done for such a large data set), as well as the manuscript. I have 2 comments/edits.

  1. For papers 633 (B/Human/IM, 28d) and 2531 (C/Mouse & Rat/IM, d10) in the supplementary information, in the table these results are defined as absolute peaking. However, there is only 1 data point after what looks to be the peak. While it does look like the 3rd point has decreased levels (supporting peaking) this is only one data point. While this reviewer acknowledges that the authors did not perform this study so that more data points and doses could be included and the equation used supports peaking, the authors should be cautious in their conclusions using papers with few data points. A good place for this to be included is where the authors comment in their discussion that with most of the data they found that they could not make a conclusion. A comment can be make that some were based off of limited data, but as a whole studies supported the peak curve conclusion.
  2. Figure 3: Please fix so that IM for Human, Rat, and Monkey wording is visible.

Author Response

Thank you for including the author’s response to reviewers to aid in review as well as track changes and highlights. The authors have made extensive changes to the supplemental material to make it much easier to read and follow (as much as can be done for such a large data set), as well as the manuscript. I have 2 comments/edits.

 We thank the reviewer for their comments, and appreciate their understanding of the difficulty in presenting a large data set.

For papers 633 (B/Human/IM, 28d) and 2531 (C/Mouse & Rat/IM, d10) in the supplementary information, in the table these results are defined as absolute peaking. However, there is only 1 data point after what looks to be the peak. While it does look like the 3rd point has decreased levels (supporting peaking) this is only one data point. While this reviewer acknowledges that the authors did not perform this study so that more data points and doses could be included and the equation used supports peaking, the authors should be cautious in their conclusions using papers with few data points. A good place for this to be included is where the authors comment in their discussion that with most of the data they found that they could not make a conclusion. A comment can be make that some were based off of limited data, but as a whole studies supported the peak curve conclusion.

We thank the reviewer for their comment. We agree that it is worth considering the possibility that, even in the cases where this modelling approach did determine evidence to support one curve shape over the other, having few data points may reduce both the specificity and sensitivity of these methods. We have added the following to the conclusion:

“Secondly, for 73% of the data, no evidence for either curve shape could be determined. A potential cause for this was that, for some of the datasets, the number of doses analysed was too few or doses too similar in magnitude. It was rare for doses to cross more than three log10, which is likely needed to see large differences. Even for datasets where evidence supporting one of the curve shapes was found, further empirical data collection may be useful to confirm the model classifications.

Figure 3: Please fix so that IM for Human, Rat, and Monkey wording is visible.

We thank the reviewer for alerting us that this was not legible on all screens. We have altered the highlighting, colouring, and format of the figure so that the wording is more visible. Please see the attachment.

Reviewer 3 Report

The authors conclude that  “Where data were informative, dose-response was approximately five times more likely to be peaking than saturating.” Thus the peaking dose response is more likely than the saturating dose response. However with the modeling techniques herein, the authors state that there is not enough evidence to discriminate between saturating and peaking dose response curves for the majority of the data sets.

In fact, it looks like the modeling techniques are not appropriate to assess the data. As this reviewer stated previously several data sets appear to fit a saturating curve, yet the models cannot discriminate between the two types of curves (see samples 936-ADHA wt Ab Week 5, 2531-gD Ab D10, 1539 AdC7PA-14dpi, 576 Ad35-PBMC, 578 Ad26-Angola, and 578 Ad26-S/G.)

There are two ways to interpret the modeling results. 1.) There is not enough evidence within the data sets to discriminate the curves. But when there is data, the peaking curve is more likely. This is the authors’ interpretation. Or 2.) The models are not able to accurately discriminate between saturating and peaking data sets, and thus the analysis is wrong. Given the evidence that numerous data sets appear to support a saturating model, yet the modeling cannot predict these curves, the analyses must be flawed and no conclusions can be drawn.

Author Response

The authors conclude that  “Where data were informative, dose-response was approximately five times more likely to be peaking than saturating.” Thus the peaking dose response is more likely than the saturating dose response. However with the modeling techniques herein, the authors state that there is not enough evidence to discriminate between saturating and peaking dose response curves for the majority of the data sets.

In fact, it looks like the modeling techniques are not appropriate to assess the data. As this reviewer stated previously several data sets appear to fit a saturating curve, yet the models cannot discriminate between the two types of curves (see samples 936-ADHA wt Ab Week 5, 2531-gD Ab D10, 1539 AdC7PA-14dpi, 576 Ad35-PBMC, 578 Ad26-Angola, and 578 Ad26-S/G.)

There are two ways to interpret the modeling results. 1.) There is not enough evidence within the data sets to discriminate the curves. But when there is data, the peaking curve is more likely. This is the authors’ interpretation. Or 2.) The models are not able to accurately discriminate between saturating and peaking data sets, and thus the analysis is wrong. Given the evidence that numerous data sets appear to support a saturating model, yet the modeling cannot predict these curves, the analyses must be flawed and no conclusions can be drawn.

We thank the reviewer for re-reviewing this work. 

It is entirely understandable that the reviewer believes from inspecting these datasets that a saturating curve shape “appear” to best describe these vaccine dose-response curves, particularly as that is the commonly held belief for the typical dose-response curve shape in the field. 

However, in our quantitative analysis it was revealed that in fact a peaking dose-response curve shape provides an equally good fit for many of the datasets, highlighting that objectively the true dose response shape cannot be distinguished from the limited empirical data available for those datasets.

We are not saying that the true dose response curve is not saturating for these datasets, just that there was not enough empirical data to objectively distinguish between the peaked or saturating. This would require new data collection at one or more dose levels.

The reviewer comments that “The models are not able to accurately discriminate between saturating and peaking data sets, and thus the analysis is wrong”. We disagree with the suggestion that because the approach was not able to discriminate dose-response curve shape for much of the data means that it was ‘wrong’. To discriminate between saturating and peaking data sets, we used an AIC cut off value of two, which is the commonly used threshold [1].

We thank the reviewer for taking the time to highlight datasets that they believe support their belief that these are saturating dose response curves. However, for each we have taken the time to detail our rationale for why objectively our results do not support the reviewer’s position (see attached pdf). 

  1. Raftery, A.E. Bayesian Model Selection in Social Research. Sociol. Methodol. 1995, 25, 111–163.
